# Recent Advances in Methods for Circulating Tumor Cell Detection

**DOI:** 10.3390/ijms24043902

**Published:** 2023-02-15

**Authors:** Monika Vidlarova, Alona Rehulkova, Pavel Stejskal, Andrea Prokopova, Hanus Slavik, Marian Hajduch, Josef Srovnal

**Affiliations:** 1Institute of Molecular and Translational Medicine, Faculty of Medicine and Dentistry, Palacky University in Olomouc, 779 00 Olomouc, Czech Republic; 2Laboratory of Experimental Medicine, University Hospital in Olomouc, 779 00 Olomouc, Czech Republic; 3Centre National de la Recherche Scientifique, Institut des Neurosciences Cellulaires et Intégratives, Université de Strasbourg, 67000 Strasbourg, France

**Keywords:** circulating tumor cells, detection, enrichment, characterization, microfluidic

## Abstract

Circulating tumor cells (CTCs) are released from primary tumors and transported through the body via blood or lymphatic vessels before settling to form micrometastases under suitable conditions. Accordingly, several studies have identified CTCs as a negative prognostic factor for survival in many types of cancer. CTCs also reflect the current heterogeneity and genetic and biological state of tumors; so, their study can provide valuable insights into tumor progression, cell senescence, and cancer dormancy. Diverse methods with differing specificity, utility, costs, and sensitivity have been developed for isolating and characterizing CTCs. Additionally, novel techniques with the potential to overcome the limitations of existing ones are being developed. This primary literature review describes the current and emerging methods for enriching, detecting, isolating, and characterizing CTCs.

## 1. Introduction

According to data compiled by the International Agency for Research on Cancer using GLOBOCAN 2020 estimates, 19.3 million new cases of cancer were diagnosed among the global population in 2020. However, despite considerable progress in the diagnosis and treatment of oncological malignancies, mortality rates remain high: cancer caused 9.9 million deaths globally in 2020 [1]. Mortality is most often caused by the emergence of distant metastases, which can develop from circulating tumor cells (CTCs). Consequently, patients at high risk of metastasis can be identified based on the detection of CTCs in blood or disseminated tumor cells (DTCs) in bone marrow [2,3].

However, the presence of CTCs/DTCs does not necessarily lead to distant metastases. Successful metastasis depends on many variables, including both the phenotypic and genotypic properties of tumor cells and the immune responses of the host organism. According to published estimates, only one in 10,000 CTCs may develop into a metastatic lesion [4,5]. Moreover, only a few CTCs are likely to reach a distant organ, survive in a dormant state, evade the immune system and systemic therapy, and eventually grow into an overt metastasis [6]. Molecular phenotypes associated with epithelial–mesenchymal transition (EMT) and stemness have been linked to increased metastatic potential and chemoresistance in CTCs [7]. Conversely, the senescent phenotype may reduce metastatic potential by inhibiting proliferation and facilitating elimination by immune cells [8]. The molecular characterization of CTCs, which will play a central role in evolving methods for CTC detection, has the potential to unravel the biology of tumor evolution and resistance and to shed new light on cancer progression at the genomic, transcriptomic, proteomic, and metabolic levels [6,9,10]. Moreover, detecting actionable variants that have not been identified in the primary tumor can help to guide treatment [11].

A sample’s CTC count typically depends strongly on the choice of detection methods and may vary from one to several thousand [12]. CTC counts can also depend strongly on the timing of blood sampling and change dynamically in ways that reflect the ongoing dissemination of the parent tumor because a CTC’s lifetime in the bloodstream is only 1 to 2 days. Another complication is that tumor heterogeneity causes considerable variation in the type and number of CTC markers in blood samples [13]. CTC detection markers commonly used for epithelial cancers include epithelial cell adhesion molecule (EpCAM) and cytokeratins (CKs). Other markers are used to characterize CTCs originating from different tumor types, including human epidermal growth factor receptor 2(Her2) and mucin 1 (MUC1) for breast cancer, prostate-specific antigen (PSA), prostate-specific membrane antigen (PSMA), androgen receptor (AR), and epidermal growth factor receptor (EGFR) for prostate cancer, and carcinoembryonic antigen (CEA) for colorectal cancer [2].

CTC counting is most commonly performed in peripheral blood samples [14]. However, new methods that exploit technological advances to detect CTCs in other body fluids are emerging. These approaches are important because of their potential to open up new sources of CTCs and advance research on CTC clusters in metastatic cancers [15]. In particular, malignant pleural and peritoneal effusions are a richer source of CTCs and CTC clusters than peripheral blood [16,17]. Accordingly, the enumeration of CTCs in cerebrospinal fluid (CSF) has been shown to enable more accurate tumor burden assessment than standard CSF cytology [18], and may thus avoid the limitations of methods based on tissue biopsy and neuroimaging [19]. Similarly, the capture of tumor cells in urine could enable the diagnosis and prognosis of bladder cancer, eliminating the need for painful endoscopy [20].

CTCs were first detected and documented by the Austrian pathologist T. R. Ashworth approximately 150 years ago [21] and are currently identified using several different methods [22,23,24,25,26]. In addition to well-established methods based on immunohistochemistry, flow cytometry, and the real-time reverse transcription polymerase chain reaction (RT-PCR), novel methods for direct detection have recently been introduced. These new techniques can increase the sensitivity of CTC detection, offer greater clinical utility than established methods, and facilitate the isolation and cultivation of live CTCs. In the first part of this review, we describe methods currently used for CTC enrichment and detection. The second part focuses on emerging methods based on rapidly evolving modern technologies, including nanomaterials, 3D printing, and artificial intelligence. 

## 2. Current Enrichment and Detection Techniques

### 2.1. Enrichment Techniques

#### 2.1.1. Morphology-Based Approaches

The diameter of CTCs (~16–20 μm) significantly exceeds that of other blood cells such as red blood cells (RBCs; ~8 μm) and white blood cells (WBCs; ~8–14 μm), and is often used to separate them [27]. Relatively simple size-based methods for capturing CTCs include isolation by size of epithelial tumor cells (ISET) [27,28,29] and techniques using a micro-electro-mechanical system (MEMS) [30].

Commercial ISET devices are available from ScreenCell (Screen Cell, Paris, France). As shown in Figure 1, filters with captured CTCs can be placed on standard glass microscopy slides for cytological analyses, or in multi-well tissue culture plates or tubes for nucleic acid or protein extraction [31]. The Screen Cell showed a 55% recovery rate and 100% specificity in blood samples spiked with the MDA-MB-231 breast cancer cell line [32].

Separation by density gradient centrifugation can also be used for CTC enrichment. Several commercial devices, kits, and reagents are available for this purpose, including the Ficoll-Hypaque (Cytiva, Marlborough, MA, USA) and OncoQuick (Hexal Gentech, Holkirchen, Germany and Greiner Bio-One, Kremsmünster, Austria) systems, which can isolate CTCs from whole blood via centrifugation in tubes with a porous barrier and a medium providing an appropriate density gradient [33]. The procedure is rapid and convenient and can provide viable cells suitable for further analysis. However, the method’s sensitivity is low. The OncoQuick showed only a 35% recovery rate in blood samples spiked with the SW-480 colon cancer cell line [34] and depends on the tumor cells’ characteristics, the centrifugation time, and the temperature. Efforts are being made to overcome these limitations [35].

Dielectrophoresis (DEP) is a relatively new and continually evolving method for isolating CTCs based on their dielectric properties. A cell’s dielectric properties (notably, its polarizability) depend on its diameter, membrane area, density, conductivity, and volume [36]. DEP can be combined with field-flow fractionation (FFF), where cells are injected into a chamber and subjected to an alternating electric field and a precisely controlled hydrodynamic flow (Figure 2) [37,38]. DEP-FFF can reportedly detect one tumor cell among 10^5^ peripheral blood mononuclear cells [39], does not require cell labeling, and allows the capture of viable cells that can be isolated and cultured. Its disadvantages include the possibility of dielectric interactions between cells and changes in their dielectric properties during prolonged storage [40]. A commercial DEP system is currently sold by Apocell under the name ApoStream (Apocell company, Houston, TX, USA) [41]. The ApoStream expressed varying recovery rates of 55–68% depending on the cancer cell lines used for spiking experiments (A549 lung cancer cell line, MDA-MB-231 triple-negative breast cancer cell line, and ASPS-1 sarcoma cell line) [42].

Several commercial cell separation systems based on microfluidic chips have also been developed in recent years. For example, ANGLE introduced the Parsortix^TM^ Cell Separation System (ANGLE, Guildford, UK), a semi-automated commercial microfluidics system with programmable fluidics and pneumatics that was designed for use with special Parsortix cassettes [43]. Additionally, in 2018, Vortex Biosciences commercialized the VTX-1 Liquid Biopsy System (Vortex Biosciences, Pleasanton, CA, USA), which uses laminar microscale vortices to isolate and enrich CTCs from whole blood based on cell size, shape, and deformability. This system achieved cell recovery rates of 69–79% in tests using breast and lung cancer cell lines [44,45]. DeNovo Sciences developed the JETTA^TM^ (Denovo Sciences, Yerevan, Armenia) microfluidics chip, which has capture chambers that can isolate CTCs based on size with up to an 83% recovery rate in tested samples spiked with cancer cell lines, but which cannot characterize them [46]. An alternative microfluidic separation technique involves the use of Dean flow fractionation (DFF), in which a blood sample and “sheath fluid” are, respectively, pumped through the outer and inner inlets of a device with a spiral microchannel, generating a centrifugal force. The RBCs and WBCs migrate along the Dean vortices while the larger CTCs are subject to strong inertial lift forces and are fractionated along the microchannel’s inner wall. This enables the continuous collection of viable CTCs [47]. The main component of the ClearCell^®^ FX System (Genomax Technology, Bangkok, Thailand), one of the first automated DFF-based systems, is a CTChip^®^ FR biochip with spiral inner and outer microchannels designed to enrich viable CTCs from whole blood [48]. The ClearCell showed a higher than 60% recovery rate in blood samples spiked with the NCI-H1650 lung cancer cell line [49].

#### 2.1.2. Immunology-Based Approaches

CTC enrichment using the magnetic-activated cell sorting system (MACS)(Miltenyi Biotec GmbH, San Jose, CA, USA) involves labeling CTCs with superparamagnetic MACS MicroBeads coated with antibodies specific for CTC surface antigens. Samples for separation are passed through a MACS Column inside a MACS Separator that contains a strong permanent magnet, causing labeled cells to be retained while unlabeled cells pass through unimpeded. The retained cells are then eluted from the column; as a result, both labeled and unlabeled cells can be isolated (Figure 3), leading to flexible and specific CTC enrichment. MACS offers high sensitivity—CTCs can be detected at concentrations as low as 1 per 10^7^ blood cells. In addition, recent updates to the system have enabled automated separation [50].

Immunomagnetic beads such as those used in MACS have several useful properties including a large surface area with many binding sites. This has been exploited to develop a range of smart solutions for isolating viable CTCs [51]. For example, the StrepTactin technology combines magnetic beads coated with mutated streptavidin molecules and a secondary antibody (anti-EpCAM, anti-EGFR, or anti-HER2) to specifically capture CTCs with 90% recovery rate using cancer cell lines. The captured cells can then be released by adding D-biotin [52]. Another smart solution is based on so-called NanoOctopus devices featuring a “head” consisting of a magnetic nanoparticle and “tentacles” composed of long single-stranded deoxyribonucleic acid (ssDNA) sequences that bind specifically to target biomarker proteins on the cell membrane. The NanoOctopus was shown to capture cells present at very low concentrations (1 to 10 cells per mL) in whole blood with a detection rate of 20–35% [53].

Another recently described system uses magnetic beads coated with a thin layer of hydrogel (MNPs@hydrogel) containing anti-EpCAM antibodies. In this case, CTCs are released from the magnetic nanoparticles by adding glutathione and are counted using flow cytometry or microscopy. A recovery rate of 98% was achieved in experiments using these beads and MCF-7 cells, with 95% of the captured cells showing good viability [54].

The cell adhesion matrix (CAM) invasion assay was specifically developed to enrich and identify invasive CTCs based on their ability to bind to, invade, and ingest a CAM, which most non-tumor and dead cells lack [55]. Here, invasive CTCs are defined as cells exhibiting CAM invasion and expressing standard epithelial markers (e.g., cytokeratins and EpCAM). Importantly, the CAM-enriched cells are viable and amenable for further analysis [55,56]. The benefits of this assay have been proven in studies on the invasiveness and tumor progenitor phenotypes of CTCs [57]. CAM invasion assay kits are sold under the Vita assay^TM^ brand (Applied DNA Sciences, Stony Brook, NY, USA). These kits include 6-well plates in which the bottom of each well is coated with a CAM layer that may be fluorescently labeled [22,58]. The methods mentioned in this section are summarized in Table 1.

### 2.2. CTC Detection Techniques

Many techniques developed for the quantitative detection of CTCs require preliminary enrichment. However, some nucleic-acid- and optical-based methods can detect CTCs without enrichment even in the presence of much larger numbers of other cells [59].

#### 2.2.1. Nucleic-Acid-Based Detection Methods

Reverse transcription polymerase chain reaction (RT-PCR) analysis has been one of the most frequently used methods for identifying CTCs [60,61]. In real-time RT-PCR, the CTC-specific messenger ribonucleic acid (mRNA) is transcribed and amplified by PCR with fluorescent dyes to quantify the exact amount of CTC-specific mRNA in the broad dynamic range [62,63]. Real-time PCR is a highly sensitive method for detecting and quantifying nucleic acids that can detect a single template molecule (and thus a single CTC) among 10^7^ normal blood cells [64]. The high specificity and sensitivity of the real-time PCR is ensured by the right selection and precise design of specific primers and probes [65]. Many tumor-specific genes and their transcripts were used as robust CTC markers (e.g., EGFR, CEA, and CKs). The combination of two or more specific markers has proven to be the most effective. Zhao et al. detected a high positive rate with either one of the three markers (EpCAM, CK19, and hMAM) in 87.8% of metastatic breast cancer patients. Simultaneously, none of the 30 healthy volunteers were positive for the detected markers [66]. However, its high sensitivity can also cause false-positive results resulting from the illegitimate expression of some CTC markers in non-cancer cells [67]. Another considerable drawback of this method is the inability to isolate, visualize, and characterize the detected CTCs.

#### 2.2.2. Cytometry-Based Detection Methods

Immunocytochemistry (ICC) enables the detection of one tumor cell among 10,000 to 100,000 non-tumor cells [68] by using fluorescently labeled monoclonal antibodies against specific tumor cell antigens in conjunction with automated imaging systems. The disadvantages of ICC include the limited number of cells that can be evaluated, the risk of cross-reactions with other epitopes, and a lower sensitivity than methods such as RT-PCR [69,70,71].

Flow cytometry (FC) enables the determination of the quantities of surface and intracellular antigens in individual cells (and thus the detection of specific kinds of cells) using monoclonal antibodies conjugated with fluorescent dyes. The most frequent target antigens for CTC detection are cytoskeletal proteins and cytokeratins [72,73]. The identified cells can be easily isolated for further analysis using a fluorescence-activated cell sorting (FACS) instrument [72]. The analysis of serial dilutions of the human breast cancer cell line SKBR-3 in blood samples of healthy donors demonstrated the detection of one CTC per 100,000 WBCs [74].

Automated digital microscopy (ADM) and fiber-optic array scanning technology (FAST) are cytometry-based techniques that involve the image analysis of ICC-labeled tumor cells. ADM has several disadvantages—notably, it requires an enrichment step and scans at very low rates (800 cells/s) [75,76]. Compared to ADM, FAST offers comparable sensitivity, greater specificity, and 500 times the scan rate while requiring no enrichment step [77]. The specificity of the FAST method was tested in the blood samples of healthy donors spiked with the colorectal cancer cell line HT-29 and showed the detection rate 1.5 × 10^−5^, and sensitivity of 98% [78]. Both FAST and ADM have proven useful for identifying very rare epithelial cells in whole blood samples after pretreatment with fluorescently labeled anti-cytokeratin antibodies [75,77].

#### 2.2.3. Microscopy-Based Detection Methods

A next-generation “liquid biopsy” technique enabling the detection and characterization of CTCs has been commercialized by RareCyte^®^ (RareCyte, Inc., Seattle, WA, USA). In this method, CTCs are isolated from whole blood, applied to a slide, stained with specific immunofluorescent dyes, and analyzed via automated microscopic imaging with an integrated fluid-coupled picking system for single-cell retrieval. Kaldjian et al. reported that the rate of successful CTC retrieval using this method was 80%–90% [25] and subsequently showed that the retrieved cells can be used for tumor characterization based on tumor-specific protein biomarkers [11].

The CytoTrack (2C A/S, Lyngby, Denmark) method also uses automated microscopic imaging for CTC detection. Cells are stained with a specific mix of immunofluorescent dyes before being placed on a glass disc, immobilized, covered with mounting medium, and scanned. The recovery rate of CytoTrack is comparable to that of CellSearch. As with RareCyte, CytoTrack allows the retrieval of single cells and the further specific molecular characterization of CTCs [79,80]. The methods mentioned in this section are summarized in Table 2.

### 2.3. Approaches Combining CTC Enrichment and Detection

The EPithelial ImmunoSPOT (EPISPOT) assay system uses antibodies to detect marker proteins specifically produced and secreted by CTCs during a 48 h cultivation period [81,82,83]. Immunospots corresponding to one marker protein-secreting cell can be counted using imaging microscopy [81]. Only viable cells are detected because dying cells secrete insufficient quantities of the marker proteins [84]. However, a cell culture facility is required and the protein used to identify CTCs must be actively secreted, shed, or released from the cells [81]. The EPISPOT reported a varying recovery rate of 37–100% depending on the dilution in blood samples spiked with the MCF-7 breast cancer cell line [85].

In the AdnaTest detection system (Qiagen, Hilden, Germany), CTCs are detected and isolated from the peripheral blood via a combination of immuno-magnetic separation and multiplex RT-PCR in order to determine the gene expression profiles of specific tumor-associated markers [86]. These assays offer specific enrichment and high sensitivity, with the ability to detect as few as 2 CTCs/10^10^ blood cells [87,88,89].

The CellSearch system (Menarini Silicon Biosystems, Castel Maggiore, Italy), which has been approved by the American Office of Food and Medical Devices (Food and Drug Administration, Silver Spring, MD, USA), was developed by Menarini Silicon Biosystems to detect CTCs in the blood of patients with metastatic breast, colorectal, or prostate cancer [90,91]. This semi-automated system involves EpCAM-based immuno-magnetic separation followed by immunofluorescence imaging and the detection of the CTCs of epithelial origin (Cluster of Differentiation (CD) 45-; EpCAM+; CK8+, 18+, and/or 19+) [92,93]. Negative selection is performed using allophycocyanin-labeled antibodies against the common leukocyte antigen CD45, while positive selection uses phycoerythrin-labeled antibodies against the common epithelial cell antigens CK8, 18, and 19. DAPI (4′,6-diamidino-2-phenylindole) is used to visualize the nuclei. This system allows CTCs to be separated from other blood cells and counted. The recovery rate was up to 80% in blood samples spiked with the SK-BR-3 breast cancer cell line [94].

The MAINTRAC method involves laser scanning cytometry after staining with EpCAM and CD45 antibodies. CTCs are identified in just two steps via automated fluorescence microscopy, which minimizes cell damage and loss [95,96]. Unlike other methods, MAINTRAC does not clean or enrich cells and detects changes in CTC counts over time [97].

Researchers at Massachusetts General Hospital developed a highly sensitive silicon CTC chip (Massachusetts General Hospital, Boston, MA, USA) to isolate viable tumor cells from whole blood. This chip has the dimensions of a microscope slide and contains 78,000 microspots coated with an EpCAM antibody [98,99]. The isolated CTCs are viable and suitable for further analysis. The second generation version of this device, termed the herringbone chip, offers higher sensitivity, with a recovery rate of 94% in spiked cancer cell line samples, as well as a greater blood volume processing capacity and more convenience [100,101] (Figure 4). Another chip-based device for CTC separation is the CTC-iChip, which combines the advantages of microfluidics and magnetism-based cell sorting. To use this device, CTCs, WBCs, and granulocytes must be immunomagnetically labeled with EpCAM, CD45, and CD15 antibodies, respectively, and after which the cells are sorted based on size. Combining the immunomagnetic labeling of target cells and WBCs with size-based separation in this way increases the sensitivity and specificity of isolation, leading to a recovery rate of 89.9% in spiked cancer cell line samples [102,103,104]. Another CTC isolation tool using chip technology is the NanoVelcro chip (UCLA, Los Angeles, CA, USA), in which anti-EpCAM antibodies are coated on silicon nanowires and cell capture is accelerated by using a chaotic mixer to generate vertical flows [105]. NanoVelcro achieves CTC recovery rates of 70% in spiked cancer cell line samples [105]. The methods mentioned in this section are summarized in Table 3.

### 2.4. Beyond the Detection—Studying the Biology of CTCs

The clinical relevance of CTCs in cancer dissemination and progression is well known [106]; however, a lot of efforts are made to go beyond CTC detection by elucidating their biology. The molecular characterization of CTCs may unravel specific actionable aberrations implicated in the tumor evolution and mechanisms of metastases [107]. However, the molecular profiling of CTCs underwent slower progress, partly due to technical challenges in isolating single cells. Immunohistochemistry, in situ hybridization, and RT-PCR have been established tools for CTC detection and cell phenotype and genome assessment for many years [6,9,108]. Additionally, methods such as digital PCR and BEAMing PCR (Beads, Emulsion, Amplification, Magnetics) for the sensitive detection of specific therapy-related mutations in CTCs have been often used [107]. Although genetic studies of CTCs can exploit the benefits of NGS, they are limited by the low yield of DNA from a single cell. Thus, methods for whole genome amplification (WGA) have been developed. Multiple-displacement amplification is the first widely used non-PCR-based WGA method with high coverage and uniformity. However, it may cause branched amplification on random locations across the genome, introducing coverage biases. An innovation occurred by the development of the MALBAC method (multiple annealing and looping-based amplification cycles) as it reduces amplification bias and reduces allelic dropout [109]. Although further innovations and new methods in the field of WGA need to be developed, some promising advances have recently been made (see Section 3.4).

## 3. Evolving Methods for CTC Detection and Characterization

Research efforts have been recently focused on developing novel and/or improved CTC detection methods by exploiting technological advances in cutting edge fields such as microfluidics and nanotechnology, as well as automation, in order to increase yields, sensitivity, and specificity as well as the potential for downstream analysis [2,110]. The development of artificial intelligence (AI) and machine learning has started a new, rapidly developing chapter of medical research with improved speed and objectivity in the detection of rare cells [111]. Microfluidic-based technologies typically separate cells by employing either internal forces such as dynamic fluid forces or external forces such as magnetic or electric fields in conjunction with cell properties such as size, density, and shape [112]. Target cells are selectively captured based on their physical properties or the presence of specific biological molecules, depending on the design of the method. Despite possible high initial costs and long set-up times, their wide-ranging capabilities have greatly facilitated the development of new CTC analysis strategies [2].

While some challenges in CTC research remain, several new or combined approaches provide the rapid and automatic capture of viable CTCs. Current efforts in CTC research also seek to go beyond enumeration to improve the accuracy of cancer diagnostics. Particular areas of interest include genotypic and phenotypic analysis of CTCs via molecular characterization, ex vivo expansion, and the investigation of crosstalk between CTCs and other cells including stromal and immune cells. Such analyses could provide unprecedented insights into the metastatic process as well as revealing new prognostic biomarkers and therapeutic targets for cancer management.

As the preceding discussion shows, several methods for CTC separation and analysis developed in the last decade have been commercialized and are sold under specific brand names. Many emerging methods for capturing, counting, and enriching CTCs use novel technologies to expand on the capabilities of these existing methods. In this section we therefore highlight technological innovations which we consider likely to be important for the close future of the field. For example, any lab equipped with a 3D printer and/or IT specialists could use them to develop new concepts or adapt existing ones for the rapid detection of intact CTCs followed by innovative downstream analyses.

### 3.1. Enrichment Techniques

#### 3.1.1. Morphology-Based Approaches

Many recently developed CTC detection methods rely on microscopic detection and image analysis. This has prompted efforts to develop machine learning algorithms to facilitate the location and classification of cells in fluorescence micrographs. Such algorithms have several input variables that typically include the intensity of each pixel in the image, the total and maximum intensity, and the intensity standard deviation, which are used to automatically evaluate each potential CTC. Wang et al. [113] used a pre-screening algorithm based on detecting the centers of cell-sized bright regions. A script was then used to normalize each image by subtracting the median and dividing it by the median absolute deviation in the pixel intensities. Normalized DAPI images are required for this step. CTCs are distinguished from the background based on local maxima where the DAPI signal and a positive marker signal (e.g., cytokeratin) are unusually bright. The next steps involve evaluating the area, perimeter, eccentricity, signal intensity, and other parameters. Finally, CTCs are distinguished from non-CTCs using machine learning algorithms implemented in MATLAB Statistics and Machine Learning Toolbox (The MathWorks, Inc., Natick, MA, USA) with pre-processing via principal component analysis [113,114].

A deep learning artificial intelligence (AI) model for multi-dimensional morphological analysis is used by the Deepcell platform (Deepcell, Inc., Menlo Park, CA, USA). This technology identifies and isolates viable cells based on morphological features. A cell suspension is loaded onto a microfluidic chip and each cell flows through the imaging area, where high-resolution bright-field images are captured for analysis using deep learning AI, which identifies CTCs based on their morphological profiles. The CTCs isolated in this way are label-free and intact and can be used to study tumor heterogeneity using molecular multi-omic techniques. The system’s capacity to detect malignant cells was evaluated using copy-number variations (CNV), targeted mutation, bulk RNA-sequencing, and single cell RNA (scRNA)sequencing analyses [115].

Some ongoing efforts seek to develop isolation techniques that do not rely on antibodies or the size of cancer cells. For example, Loeian et al. [23] developed a nanotube CTC chip that combines carbon nanotube surfaces with batch manufacturing techniques in a 76-element microarray for antigen- and size-independent capture and the isolation of tumor-derived epithelial cells. Their approach is based on the hypothesis that CTCs will preferentially adhere to a carbon nanotube surface. This hypothesis is supported by electron micrographs showing that cells in contact with a nanotube surface undergo morphological changes and bind to the nanotube matrix via filaments extending from the main body of the cell. The first step in CTC capture using this system involves lysing RBCs from an 8.5 mL blood sample. WBCs are then pelleted via centrifugation and CTCs are enriched by capturing them on the surfaces of chip-bound nanotubes. Finally, the attached CTCs are immunostained on-chip using DAPI and antibodies against cell surface antigens including CKs 8/18, Her2, and EGFR for CTC identification as well as anti-CD45 antibodies for WBC identification. CTC recovery rates of 87–100% in spiked cancer cell lines have been achieved in this way [23].

Another emerging method is based on the observation that non-adherent and circulating tumor cells produce tubulin-based protrusions or microtentacles (McTNs) when detached from the extracellular matrix, which helps tumor cells form clusters during metastasis. Ju et al. [24] took advantage of this behavior by developing a microfluidic device with a recovery rate of 98% in spiked cancer cell lines called TetherChip, where CTCs are captured on the thermal-crosslinked polyelectrolyte multilayer nanosurface with a terminal lipid layer. This allows non-adherent CTCs to be chemically fixed and stabilized for over 6 months without alteration of their morphology or phenotype, which is particularly important when studying free-floating tumor cells [24] (Figure 5).

#### 3.1.2. Immunology-Based Approaches

The CellCollector sold by GILUPI GmbH is a device consisting of an EpCAM-coated gold wire that is inserted into a patient’s vein via a cannula to perform whole blood volume analysis [116] (Figure 6). During trials in patients with colorectal cancer, its detection rate was compared to that of the CellSearch System. Although a larger volume of blood passed through the CellCollector, its CTC yield and sensitivity did not differ significantly from those achieved with CellSearch [117].

Other effective devices for CTC capture and recovery have been developed using different modern technologies. For example, a microchip with a 3D conductive scaffold made from porous polydimethylsiloxane with immobilized gold nanotubes (Au-NTs) coated with an anti-EpCAM antibody achieved a recovery rate of 92.8% in spiked cancer cell lines with 93.9% viability. The scaffold’s 3D structure induces chaotic migration and promotes interactions between CTCs and the substrate, facilitating their capture [118]. Moreover, 84.3% of the captured cells could be reversibly released via electrical stimulation of the scaffold. This system is also capable of CTC cluster enrichment, but clusters are released with lower efficiency than single cells, possibly because they tend to be recaptured.

A general advantage of such 3D nanostructured systems is that they have large binding surfaces and thus provide many binding sites for CTC capture. This is exemplified by a novel 3D Zn(OH)F/ZnO nanoforest array that was synthesized in a glass capillary, causing the capillary’s inner walls to be coated with Zn(OH)F nanowires. The lateral branches of these nanowires were conjugated to an anti-EpCAM antibody, creating a very large surface area for interaction cellular filopodia which allowed CTCs to be captured with approximately 90% efficiency [119].

Chen et al. [120] developed a novel 3D printed functionalized device with a 2 cm long 3D printed channel whose inner surface was functionalized with an anti-EpCAM antibody. In tests using three human EpCAM-positive cell lines (breast cancer [MCF-7], colon cancer [SW-480], and prostate cancer [PC3]) and one EpCAM-negative cell line (kidney cancer [293T]), this device achieved a CTC capture efficiency above 90% [120].

### 3.2. Detection Techniques

Epic Science (Epic Sciences, Inc., San Diego, CA, USA) introduced a novel and extremely sensitive platform with a recovery rate of up to 88% in blood samples spiked with cancer cell lines for the simultaneous detection and characterization of rare CTCs, allowing the detection of 1 CTC in up to 50 billion cells [121,122,123]. Uniquely, this platform facilitates the analysis of all nucleated cells in a blood sample to not only identify CTCs but also assess their genetic mutations and deviations in protein expression. Epic’s platform can thus determine the number of cancer cells in a sample as well as each cell’s expression of specific biomarkers and their subcellular localization. It can also provide information on the cells’ genomic status and cancer type, and by extension the heterogeneity and clonality of the patient’s cancer [124].

A new approach combines artificial intelligence with nanoarray technology to detect both cancer cells and volatile organic compounds (VOCs) from cancer cells and their microenvironment. Blood samples are analyzed using an array of chemiresistors consisting of molecularly capped spherical gold nanoparticles, 2D random networks of single-walled carbon nanotubes capped with different organic compounds, and polymeric composites. This allows the analysis to combine signals originating from multiple sensors for different chemicals, giving high detection sensitivity. Specific VOCs associated with single cancer cells are then analyzed via gas chromatography–mass spectrometry [125]. Experiments on mouse models showed that the nanoarray achieved over 81% sensitivity and 80% specificity for the early detection of breast, ovarian, and prostate cancer xenografts, as well as 100% sensitivity and over 88% specificity for detecting metastasis [126].

### 3.3. Approaches Combining CTC Enrichment and Detection

Modern 3D printing technologies offer another attractive direction for future work on CTC capture and characterization. Chu et al. [127] used 3D printing to create a microfluidic device consisting of two parts: a multilayered immunoaffinity section for leukocyte capture and a filtration section. Most WBCs are captured in the device’s immunocapture channels but RBCs, platelets, and all nucleated cells (including residual WBCs) migrate to a commercially available membrane micropore filter that retains nucleated cells while releasing RBCs and platelets. The membrane filter is removable, allowing tumor cells to be analyzed using a microscope or stained on-chip. Experiments with prostate, breast, or ovarian cancer cell lines yielded CTC recovery rates of around 90% [127].

The CTCelect system (Fraunhofer Institute for Microengineering and Microsystems IMM, Mainz, Germany) is a new fully automated cancer profiling device that combines immunomagnetic CTC enrichment with microfluidic sorting of fluorescence-activated cells [123,128]. Single cells are dispensed into a 96-well plate in microliter droplets and analyzed via fluorescence microscopy, yielding CTC recovery rates of 72% from whole blood samples spiked with the cancer cell line (MCF7). RNA can then be extracted from the isolated CTCs for analysis via real-time PCR [129].

Another notable device using multiple CTC separation technologies is the VyCAP system (VyCap B.V., Enschede, Netherlands), which combines size-based filtration with automated imaging and reportedly achieves recovery rates of 65–79% for epithelial and mesenchymal human cancer cell lines [130,131]. Cells are filtered through a microsieve filter chip with a 1 µm thick silicon nitride filter membrane that has 160,000 pores with diameters of 5 ± 0.2 µm. Because the membrane is so thin, filtration is performed at low pressure, which minimizes damage to the captured cells. After filtration, captured CTCs are labeled using Cellstainer and automatically enumerated with the VyCap Imaging system. The system’s standard CTC counting protocol is based on DNA+, CK+, CD16-, and CD45- labeling, but other cancer-specific labels can be used such as MUC-1 and PDL-1.

The MyCTC microfluidic chip was recently introduced to facilitate drug response prediction in patients with advanced cancer. It has a polydimethylsiloxane upper layer and a bottom layer housing microfluidic structures made of a rigid cyclic olefin copolymer. The chip is divided into two sections—one for CTC capture and culture, and another for drug screening. Captured cells are thus cultivated and then transferred to drug screening chambers. This allows cancer cells to be isolated from whole blood or other body fluids without pre-processing or reliance on labels or antibodies. Single CTCs and CTC clusters are captured with recovery rates of 95–98% and 97–99% in spiked cancer cell line samples, respectively [132].

Xu et al. developed an in situ strategy that combines size- and deformability-based microfiltration with CTC capture using nanoparticle probes targeting folic acid on the cell surface [133]. Another combined system using nanoparticle probes was introduced by Jia et al., who used magnetic nanoparticles coated with a peptide targeting N-cadherin to capture CTCs that escape detection by EMT. The captured cells were then processed using a microfluidic chip. It was shown that this system can successfully capture viable mesenchymal cells that can subsequently be analyzed via RNA sequencing [134].

Finally, an approach using a so-called stereo acoustic streaming tunnel has been used to capture cells from whole blood. The cells are first separated based on their physical properties under the influence of ultrahigh-frequency bulk acoustic waves and then analyzed via immunofluorescence, leading to excellent separation efficiencies. This approach was originally developed for single cell manipulation but could be exploited in novel CTC detection and in situ characterization methods [135]. The methods mentioned in this section are summarized in Table 4.

### 3.4. Molecular Characterization Will Be Integral to Future Methods for CTC Detection

Recently, innovations in molecular biology and techniques for isolating and manipulating single cells have facilitated more comprehensive analyses of CTCs [10,108]. The genomic profiling of CTCs using targeted next-generation sequencing (NGS), microarrays, and competitive genome hybridization arrays requires precise whole genome amplification (WGA) of DNA extracted from a single cell [9]. Several PCR-based and non-PCR-based WGA methods have been developed for this purpose. However, in vitro artifacts, reaction uniformity, and allelic drop-out continue to present significant challenges in WGA; so, the method to be used must be chosen based on the desired experimental outcomes [136]. Novel methods such as primary template-directed amplification (PTA) [137] and linear amplification via transposon insertion (LIANTI) [138] have recently emerged to address some of its limitations.

Few large-cohort studies involving CTC characterization have been reported, and there is a need for additional studies using recently introduced tools such as third-generation sequencing technologies to support and enhance the current understanding of their biology and role in metastasis [108]. However, the genomic and transcriptomic profiling of CTCs has already yielded valuable insights. For example, single-cell whole genome bisulfate DNA methylation profiling revealed selective hypomethylation of genome binding sites for transcription factors promoting stemness and proliferation in CTC clusters [139]. Furthermore, CTC copy number aberration profiles were exploited in the stratification of chemosensitive and chemoresistive patients [140]. Additionally, whole-genome sequencing of CTCs revealed a 91% mutation overlap with metastases detected 10 months after sequencing [141].

Single-cell transcriptome profiling of CTCs presents several challenges arising from the dynamic nature of the transcriptome and the low stability of RNA [142]. Additionally, stabilizing agents in specimen collection tubes and sample handling procedures may induce transcriptomic changes and impact RNA quality [143]. Nevertheless, transcriptome profiling of CTCs has attracted considerable interest and new methods for this purpose have been introduced. A notable example is Hydro-Seq, which enables scRNA sequencing and identification of stemness and EMT markers [144]. Other studies have showed that the progress of cancer therapy can be monitored with high predictive accuracy by using a combination of microfluidics and digital PCR to quantify CTC-derived estrogen receptors and to thereby characterize CTC signatures [145,146]. Finally, sophisticated methods combining scRNA sequencing with other approaches such as drop-seq, smart-seq, CITE-seq, ChIP-seq, and ATAC-seq can characterize CTCs with single-cell resolution [144,147,148] but must be coupled with an appropriate CTC capture method.

Single-cell proteomics and metabolomics are emerging approaches for CTC characterization and can provide otherwise inaccessible information on their properties [10]. However, given the dynamic diversity of the cellular proteome and metabolome, CTC profiling with single-cell resolution remains challenging. These problems may be overcome by mass spectrometric methods, which can achieve very low detection limits, and microfluidic methods which facilitate the isolation and manipulation of large numbers of cells [10,149]. A variety of novel techniques using such methods are therefore emerging, including a microfluidic Western blot for an individual CTC protein panel that provides a deeper understanding of CTC biology [150]; an oil-air-droplet chip enabling single-cell treatment and liquid chromatography-mass spectrometry (LC-MS) analysis with minimal sample loss [151]; and a technique that combines droplet-generating machinery with FACS and NGS or LC-MS secretome analysis to provide information on the genotype and phenotype of single cells [152].

The development of CTC-derived experimental models such as cell lines, cell-line-derived xenografts, and patient-derived xenografts will facilitate single-cell research on CTCs [6,153] and could be used for preclinical drug screening in conjunction with genomic, transcriptomic, proteomic, and metabolomic profiling. However, the establishment of CTC-derived experimental models remains challenging and further advances are needed in this area [153].

## 4. Conclusions

Despite advances in diagnosis and treatment, cancer metastases are still some of the most often causes of death worldwide [111]. CTCs are considered to be precursors of metastases and their capture and analysis in body fluids, typically blood, may provide insights into the metastatic process and identify new biomarkers and therapeutic targets. A variety of methods for detecting and characterizing CTCs have been developed and shown to provide information that is valuable for early cancer detection and follow-up. However, these methods differ in their sensitivity and specificity, creating a risk of both false-negative and false-positive results.

The rapid development in fields such as microfluidics, nanotechnology, and computational methods, as well as molecular biology, has facilitated the innovation of CTC analyses. Thus, CTC detection and analysis methods have been improved significantly in terms of accuracy, sensitivity, specificity, and the capacity of detection. The automation and quick adoption of novel materials should significantly reduce current high initial costs and facilitate their use. Therefore, new microfluidic approaches for CTC detection and isolation have been developed which do not rely on labeling and can be combined with advanced imaging analyses. These approaches are evolving rapidly and could potentially be made suitable for routine point-of-care use while also providing the ability to isolate viable CTCs for downstream multi-omic analyses that would provide deeper insights into the metastatic process and thus facilitate treatment [154].

However, whether further improvements in sensitivity will translate to improvements in clinical outcomes or simply complicate the interpretation of results remains to be determined [155,156]. Ideally, research on CTC-specific markers and the combined analysis of CTCs with other biomarkers such as cell-free DNA or exosomes will ultimately enable the clinical use of CTCs as a “liquid biopsy” which provide detailed information on a patient’s cancer and treatment [154].

## Figures and Tables

**Figure 1 ijms-24-03902-f001:**
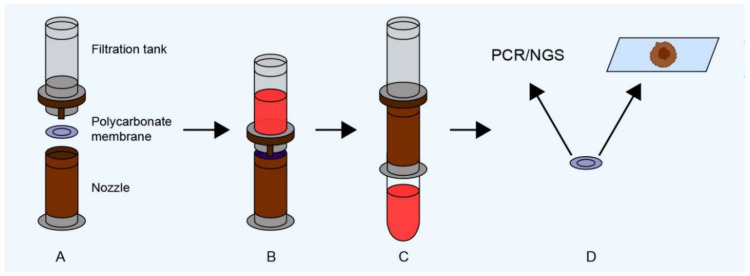
Schematic depiction of an ISET-based system for CTC enrichment. The filtration device, whose components are shown in (**A**), is used to collect blood (**B**), which is filtered through a membrane (**C**). The auxiliary components are then removed to enable further analysis of the membrane-bound CTCs (**D**) (polymerase chain reaction, PCR; next-generation sequencing, NGS).

**Figure 2 ijms-24-03902-f002:**
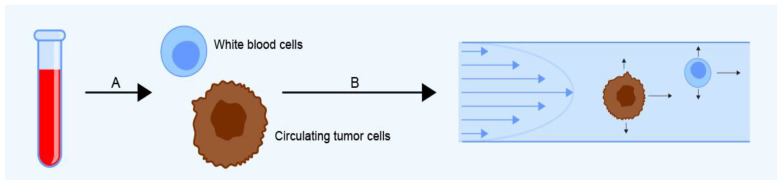
The dielectrophoretic field-flow fractionation (DEP-FFF) method allows CTCs to be isolated from blood samples (**A**) by separation in a dielectrophoretic chamber (**B**) under the influence of hydrodynamic lift and levitation forces (↑), gravitational and sedimentation forces (↓), and the fluid velocity (→).

**Figure 3 ijms-24-03902-f003:**
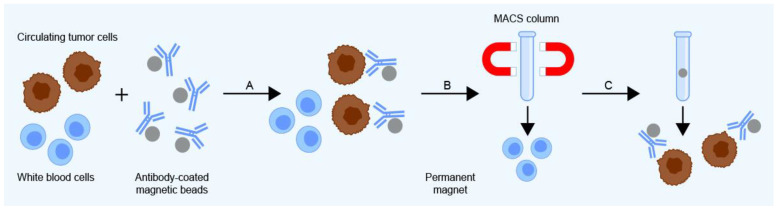
The magnetic-activated cell sorting (MACS) system uses antibody-coated magnetic beads to capture CTCs (**A**). The separator’s magnetic field causes labeled cells to be retained on the column while unlabeled cells pass through unimpeded (**B**), after which the labeled CTCs are released (**C**).

**Figure 4 ijms-24-03902-f004:**
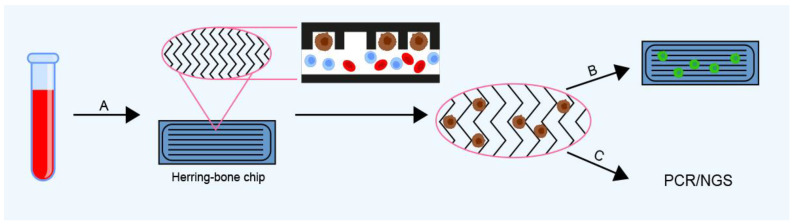
Herringbone chips (HB chips) capture CTCs directly from whole blood (**A**) in herringbone-etched microchannels, differentiating them from blood cells based on their size and mobility. Captured cells can be directly stained and enumerated (**B**) on the chip using conjugated antibodies and/or (**C**) washed out from the chip for further analysis (polymerase chain reaction, PCR; next-generation sequencing, NGS).

**Figure 5 ijms-24-03902-f005:**
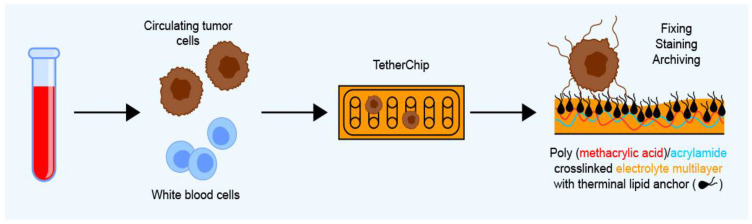
The TetherChip CTC capture system relies on the high affinity of CTC microtentacles for a crosslinked polyelectrolyte multilayer formed by thermal imidization.

**Figure 6 ijms-24-03902-f006:**
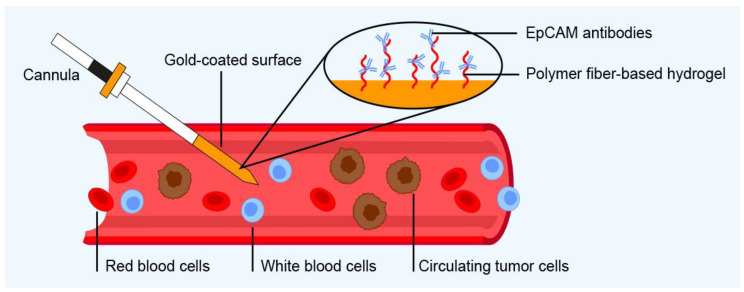
Operating principles of the GILUPI CellCollector for in vivo CTC enrichment on a functionalized surface. CTCs are captured by antibodies immobilized on a hydrogel and can be released from the surface for analysis after removing the cannula from the vein.

**Table 1 ijms-24-03902-t001:** CTC enrichment techniques.

CTC Enrichment Techniques								
	Name	Commercially Available Formats or Providers	Mode of Enrichment	Antibodies	Material	Advantages	Disadvantages	References
Morphology-based approaches	Isolation by size of epithelial tumor cells (ISET)	Screen Cell (Screen Cell, Paris, France), CTCBIOPSY^®^ (YZYBIO Company, Wuhan, China)	Filtration by size through a polycarbonate membrane with 8-µm diameter cylindrical pores	-	Blood or other biological fluid	Simplicity, high sensitivity, possibility to detect CTCs directly on membrane or isolate them. Applicable to all tumor types; low price; fast	Results may be affected by the morphological variability of CTCs, possibly leading to false-negative responses	[28,29]
	Micro-electro-mechanical system (MEMS)	-	Filtration by size through a parylene membrane with 10-µm diameter pores	-	Blood	Simplicity, high sensitivity, possibility to detect CTCs directly on membrane or isolate them. Applicable to all tumor types	Results may be affected by the morphological variability of CTCs, possibly leading to false-negative responses	[30]
	Ficoll density gradient	Ficoll-Hypaque (Cytiva, Marlborough, MA, USA), OncoQuick Assay (Hexal Gentech, Holkirchen, Germany, and Greiner Bio-One, Kremsmünster, Austria)	Density gradient centrifugation	-	Blood or bone marrow	Rapid and convenient method, provides viable cells, low price	Sensitivity is low and depends on tumor characteristics, centrifugation time, and temperature	[35]
	Dielectrophoretic field-flow fractionation (DEP-FFF)	ApoStream^®^ (Apocell company, Houston, TX, USA), Parsortix^TM^ Cell Separation System (ANGLE, Guildford, UK), VTX-1 Liquid Biopsy System (Vortex Biosciences, Pleasanton, CA, USA), JETTA^TM^ (Denovo Sciences, Yerevan, Armenia)	Separation based on the dielectric characteristics of CTCs combined with field-flow fractionation	-	Blood	Label-free, possibility to obtain viable cells that can be isolated and cultured, short processing time (~2.5 mL/h)	Possibility of dielectric interactions between cells and changes in their dielectric properties during prolonged storage	[37,40,43,44,46]
	Dean flow fractionation (DFF)	ClearCell^®^ FX System (Genomax Technology, Bangkok, Thailand)	Microfluidic separation based on centrifugal force	-	Blood	Possibility to continuously collect viable CTCs, short processing time (36 mL/h)	Less efficient for small CTCs	[48]
Immunology-based approaches	Magnetic-activated cell sorting (MACS)	MACS (Miltenyi Biotec, San Jose, CA, USA)	Capture by immuno-labeled magnetic microbeads using superparamagnetic nanoparticles and columns	Cytokeratin, EpCAM, EGFR, and HER2	Blood or bone marrow	High sensitivity, enables automated separation	Expensive	[47]
	Cell/collagen adhesion matrix (CAM) invasion assay	Vita Assay^TM^ (Applied DNA Sciences, Stony Brook, NY, USA)	Based on CTCs’ ability to bind, invade, and ingest a CAM and express biomarkers	EpCAM, Epithelial specific antigen (ESA), and pan-CK (CKs 4, 5, 6, 8, 10, 13, and 18)	Blood	Enrichment of viable cells, which can be used to determine invasiveness and tumor progenitor phenotypes of CTCs	CAM may be present in the blood of healthy references as well as cancer cases; isolation step requires more than 12 h	[22,55]

**Table 2 ijms-24-03902-t002:** CTC detection techniques.

CTC Detection Techniques								
	Name	Commercially Available Formats or Providers	Mode of Detection	Antibodies	Material	Advantages	Disadvantages	References
Nucleic-acid-based detection	Reverse transcriptase polymerase chain reaction (RT-PCR)	Range of assays for selected diagnoses	Reverse transcription of CTC-specific mRNA to complementary DNA (cDNA) followed by PCR-amplification of cDNA	-	Blood, bone marrow, tissue, and other biological samples	High sensitivity	RNA instability, illegitimate expression, and false positivity; does not allow isolation of viable cells	[2,62,64]
Cytometry-based detection	Immunocytochemistry (ICC)	Range of assays for selected diagnoses	Antibody staining of tumor-specific antigens	Chosen based on proteins expressed in primary tumors	Blood, bone marrow, tissue, and other biological samples	Can be conjugated with automated imaging system	Limited number of cells evaluated, risks of cross-reactions with other epitopes, low sensitivity	[70,71]
	Flow cytometry (FC)	Range of assays for selected diagnoses	Quantification of surface and intracellular antigens using antibodies conjugated with fluorescent dye	Chosen based on proteins expressed in primary tumors	Blood, bone marrow, tissue, and other biological samples	Ability to measure multiple parameters of large numbers of cells relatively quickly. Cells can be isolated for further analysis	Low sensitivity, time-consuming	[72]
	Automated digital microscopy (ADM)	-	Fluorescence microscopy and robotic motion control system to automate imaging	Antibodies against tumor-specific biomarkers	Blood	Identification of very rare epithelial cells in whole blood samples	Enrichment step needed, long exposure time (800 cells/s)	[75,77]
	Fiber-optic array scanning technology (FAST)	FASTcell™ (SRI International, Menlo Park, CA, USA)	Image analysis of immunocytochemically labeled tumor cells	Antibodies to tumor-specific biomarkers	Blood	Does not require enrichment step	Special type of cytometer needed	[75,76,77]
Microscopy-based detection	Fluorescence microscopy	Range of assays for selected diagnoses, e.g., CytoTrack^®^ (Cytotrack Aps, Lyngby, Denmark), RareCyte^®^ (RareCyte, Inc., Seattle, WA, USA)	Optical microscopic examination of cells stained immunologically	Based on genes expressed in primary tumors, e.g., Anti-EpCAM and cytokeratins	Blood, bone marrow, tissue, other biological samples	Enrichment-free; automated microscopic imaging system; possibility of single-cell retrieval and further molecular characterization of CTCs; can be used for non-epithelial cells	Limited observation time, manual assessment necessary; long processing time; isolated cells are not viable	[11,25,79]

**Table 3 ijms-24-03902-t003:** Approaches combining CTC enrichment and detection.

Approaches Combining CTC Enrichment and Detection								
Name	Commercially Available Formats or Providers	Mode of Enrichment	Mode of Detection	Antibodies	Material	Advantages	Disadvantages	References
EPithelial ImmunoSPOT assay (EPISPOT)	-	Negative selection using anti-CD45 immuno-magnetic beads, culture in plates pre-coated with antibodies to capture secreted protein of interest	Secreted protein spots are detected via immunological techniques and counted	Cathepsin D, MUC1, CK19, PSA	Blood	Detects only viable cells	The protein used to identify CTCs must be actively secreted, shed, or released from cells	[81,82,83]
AdnaTest	AdnaTest (Qiagen, Hilden, Germany)	Immuno-magnetic separation (AdnaTest Select)	Multiplex RT-PCR (AdnaTest Detect)	MUC-1, HER-2, EpCAM, CEA, EGFR, PSA, PSMA, Aldehyde dehydrogenase 1 (ALDH1)	Blood	Specific enrichment and high sensitivity	Not automated; processing time of 5 h; expensive	[86,89]
CellSearch system	CellSearch (Menarini Silicon Biosystems, Castel Maggiore, Italy)	Immuno-magnetic separation	Flow cytometry and immunofluorescence imaging	EpCAM, CKs 8, 18, 19	Blood	High sensitivity, specificity, and reproducibility; semi-automated; FDA approved	Low sensitivity for cells with low EpCAM expression	[90,92,93]
MAINTRAC	-	Red blood cell lysis and centrifugation	Laser scanning cytometry, automated fluorescence microscopy	EpCAM	Blood	Does not clean or enrich cells, which minimizes cell damage and loss	Cannot be used for early diagnosis	[95,96,97]
CTC-Chip	CTC-Chip, CTC-iChip (Massachusetts General Hospital, Boston, MA, USA), NanoVelcro chip (UCLA, Los Angeles, CA, USA)	Microfluidic separation on silicon chip microposts with EpCAM antibodies	Cytokeratin antibodies and DAPI	Cytokeratin	Blood	High sensitivity; isolates viable CTCs; short processing time	Does not detect CTC-WBC clusters	[98,102,104,105]

**Table 4 ijms-24-03902-t004:** Evolving methods for CTC detection and characterization.

Evolving Methods for CTC Detection and Characterization										
		Name	Commercially Available Formats or Providers	Mode of Enrichment	Mode of Detection	Antibodies	Material	Advantages	Disadvantages	References
Enrichment techniques	Morphology-based approaches	Deepcell	Deepcell platform (Deepcell, Inc., Menlo Park, CA, USA)	Isolation of viable cells based on morphological distinction	Images are analyzed using deep learning AI	-	Blood and other body fluids	Permits cluster analysis and further molecular characterization of CTCs	Not clinically validated	[115]
		Nanotube-CTC-chip	-	CTCs adhere to a carbon nanotube surface via filaments extending from the main body of the cell	CTCs are immunostained on-chip and analyzed using automated fluorescence microscopy	DAPI, CKs 8/18, Her2, EGFR, and anti-CD45	Blood	Antigen- and size-independent capture	RBC lysis is necessary	[23]
		TetherChip	-	CTCs are captured based on the affinity of their microtentacles for a polyelectrolyte multilayer	Immunofluorescence staining with Hoechst, WGA, and GFP followed by fluorescence microscopy analysis	-	Blood	Preserves microtentacle structure after fixation and isolation from blood; enables testing of functional phenotypes in CTCs	Only tested on cell lines at the time of writing	[24]
	Immunology-based approaches	GILUPI CellCollector	GILUPI CellCollector (GILUPI GmbH, Potsdam, Germany)	CTCs captured by antibodies immobilized on a hydrogel	Immunofluorescence staining and molecular analysis (e.g., PCR, sequencing, gene expression analysis)	EpCAM	Blood	Enriches CTCs directly from bloodstream rather than volume-limited blood samples; enrichment time is 30 min	Used only for enrichment of CTCs directly from patient’s bloodstream	[116,117]
		3D conductive scaffold microchip	-	CTCs are captured on a 3D conductive scaffold made from porous polydimethylsiloxane with immobilized gold nanotubes (Au-NT) coated with an anti-EpCAM antibody	Immunocytochemistry using FITC-CK, PE-CD45, and DAPI	EpCAM	Blood	Captured cells can be reversibly released with high viability; high sensitivity	CTC clusters released less efficiently than single CTCs because of re-capture by the 3D scaffold	[118]
		3D nanoforest array	-	Cellular filopodia of CTCs interact with lateral branches of Zn(OH)F nanowires conjugated to an anti-EpCAM antibody	Immunofluorescence staining and fluorescence microscopy analysis	EpCAM, CD45, CK	Blood	Large binding surfaces provide many binding sites for CTC capture	Only tested on cell lines at the time of writing	[119]
		3D-printed functionalized device	-	3D-printed channel whose inner surface was functionalized with anti-EpCAM	Confocal laser scanning microscopy	EpCAM	Blood	Microfluidic device with a large binding surface area	Only tested on cell lines at the time of writing	[120]
Detection techniques		Epic Sciences	Epic Sciences (Epic Sciences, Inc., San Diego, CA, USA)	-	Pyxis™—whole slide fluorescent scanner	Cytokeratin, CD45, DAPI, and specific antibodies	Blood	Enrichment-free; cancer profiling combining CTC technology with circulating tumor DNA (ctDNA) and immune cell analysis	Samples must be sent to the company for analysis, only for prostate and breast cancer	[121,122,124]
		AI nanoarray	-	Detects both cancer cells and VOCs from cancer cells and their microenvironment	Gas chromatography linked with mass spectrometry	-	Blood	High sensitivity and specificity for early detection	Only tested on cell lines and a mouse model at the time of writing	[125,126]
Approaches combining CTC enrichment and detection		3D-printed microfluidic device	-	WBCs are captured in the device’s immunocapture channels; RBCs, platelets, and all nucleated cells migrate to a membrane micropore filter	CTCs are immunostained on-chip and analyzed using fluorescence microscopy	CD45	Blood	Label-free negative depletion of CTCs; isolation of very small CTCs	Only tested on cell lines at the time of writing	[127]
		CTCelect	CTCelect system (Fraunhofer Institute for Microengineering and Microsystems, IMM, Mainz, Germany)	Combines immunomagnetic enrichment with microfluidic sorting of fluorescence-activated cells	Fluorescence microscopy	EpCAM	Blood	Fully automated; permits further molecular characterization of CTCs	Captures only single cells, not clusters	[128,129]
		VyCAP	VyCAP technology (VyCap B.V., Enschede, The Netherlands)	Size-based filtration through a microsieve filter chip	Fluorescence microscopy with automated imaging system	CK, CD16, and CD45. Other cancer-specific labels can also be used (e.g., MUC-1, PDL-1)	Blood	Fully automated; filtration under low pressure, which minimizes damage to captured cells	Not clinically validated	[130,131]
		MyCTC chip	-	CTCs are captured on microfluidic chip with a polydimethylsiloxane upper layer and a rigid cyclic olefin copolymer underlayer	Cultivation of captured CTCs	-	Blood or other body fluids	Label- and antigen-free; captures clusters with high efficiency	Not clinically validated	[132]

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
