# Peer review of "Recent Advances in Methods for Circulating Tumor Cell Detection"

_ijms, 2023, doi:10.3390/ijms24043902_

Round 1

Reviewer 1 Report

This work is well explained to help practical researchers who deal with CTC understand the history. Unfortunately, however,

much of the information researchers need to conduct real-world experiments is missing. Therefore It would be added  such essential informations into one table.

Would you mind adding a table showing actual CTC havesting rate, time required and cost we should pay for?

Reviewer 2 Report

Manuscript ID: ijms-2127766

Title: Current and future methods for detecting circulating tumor cells

Vidlarova et al.

Circulating tumor cells (CTCs) can reflect the current state of a tumor and therefor pose a valuable prognostic tool in cancer treatment. In their review Vidlarova et al. give an overview of the currently available methods for enrichment, detection and characterization of CTCs and give an outlook on future techniques.

While the topic of CTC detection is surely relevant to the International Journal of Molecular Sciences, the review does not provide significantly new insights into the current methodology and remains rather superficial. The introduction lacks very basic information on CTCs like common markers. This information however is crucial to understanding the methods described in the following sections. The provided tables contain valuable information, but they are too packed and hardly readable. While the figures are very nicely designed, some of the descriptions do not match with the figures content and need to be revised (see comments). Overall the review includes the most common techniques and also gives insights into more recently developed methods. However, the level of detail with which the different methods are described varies greatly. It would be very helpful to include efficiency data and examples from application for all techniques as it is done for some. The review contains a section specifically dedicated to “future methods for CTC characterization”, but it remains unclear how these methods were chosen to be “the future” of the field. Overall the manuscripts sections need to be restructured to improve the differentiation of CTC enrichment, detection and characterization. It also remains questionable how much new information can be deducted from the given review, as there are already numerous reviews published that mention more or less the same methods as described here (e.g. A Review of Circulating Tumour Cell Enrichment Technologies, Rushton et al. 2021; Circulating Tumor Cell Enrichment Technologies, Boja et al. 2020; Strategies for enrichment of circulating tumor cells, Li et al. 2020). Perhaps it would be more beneficial for to focus on the newly developed techniques and describe those in greater detail, as the common methods have already been reviewed so often.

In summary, I am unable to recommend acceptance of the review article for publication in the International Journal of Molecular Sciences in its current form and suggest „rebranding“ it to review new developments and advances in CTC enrichment, detection and characterization.

Comments:

       Introduction

â—¦      31-32: “… distant metastases that develop from CTCs” very strong claim, CTCs are not the only source for metastases

â—¦      32-34: “Consequently, patients at high risk of metastasis can be identified based on the detection of CTCs in blood or disseminated tumor cells (DTCs) in bone marrow” What about other body fluids (urine, saliva etc)

â—¦      Generally, lacks basic information about CTCs like common markers, mode of dissemination, EMT/MET, stemness, some background on the prognostic value in different cancers etc

â—¦      Does not mention CTC characterization, only enrichment and detection

       Section 2. Enrichment and detection of CTCs

â—¦      the section only contains 6 lines of text, which only gives very brief information. I would suggest deleting this section and including the given information and tables in the introduction and/or following sections

â—¦      Tables 1-4

â–ª      table format is very hard to read, tables need to be restructured in order to improve readability

â–ª      Instead of having columns for quantification, living cells, histology and characteristics it would be more reader friendly to list advantages and disadvantages of the methods, that include the information on possibility for quantification etc

       Section 3. Enrichment techniques

â—¦      Include table 1 here

â—¦      Figure 2: The description does not match the figure (letters given in text versus figure)

â—¦      Figure 3: again, description and figure don’t align too well, consider placing the letters with the actual image not on arrows to make the workflow more comprehensible

â—¦      Presentation of the NanoOctopus technique (including own figure) even though efficiency is only 20-35%? Seems unproportioned to give so much attention to a technique with such low efficiency

       Section 3.3 CTC detection techniques should be labeled section 4, separating enrichment and detection techniques

â—¦      Include table 2 here

       Sections 3.4 and 3.5

â—¦      Quite detailed description of RT-PCR and flow cytometry, as these are rather common methods a general explanation of the principle seems unnecessary, especially compared to the rather shallow explanations concerning some of the actual CTC related techniques

       Section 3.6 Approaches combining enrichment and detection of CTCs accordingly should be labeled section 5

â—¦      Include table 3 here

â—¦      250-260: explanation of immunomagnetic separation and RT-PCR again for the Adna test. Both methods have been described before, hence it is unnecessary 

       Section 3.7 Approaches combining detection and molecular characterization of CTCs accordingly should be labeled section 6

â—¦      Include table 4 here

â—¦      Comparable to RareCyte and CytoTrack there is also the CTCelect system (Stiefel et al. 2022) that provides immunomagnetic enrichment and single cell dispersion 

       Section 3.8 Molecular characterization of CTCs with single-cell resolution accordingly should be labeled section 7

â—¦      The whole section remains very superficial and gives little insights on the actual methods used to characterize CTCs

â—¦      As the title of the paper refers to detection (and prior enrichment) of CTCs the authors may consider leaving out this section/including the information in previous sections, otherwise it needs to be extended

       Section 4 Future methods for CTC characterization

â—¦      The presented methods seem more focused on enrichment and detection than characterization

â—¦      How are these methods “the future” compared to the previously described ones? Just based on being more recently developed? Cutoff unclear!

â—¦      What are the advantages/disadvantages compared to the „standard“ methods?

       Conclusion

â—¦      Very brief!

â—¦      Highly critical of EpCAM based enrichment, even though it remains the gold standard of CTC isolation and many of the “future” methods described also rely on EpCAM

       Generally, more examples from application of the different methods would be helpful and what conditions (cancer type, fluid, volume, markers) were used

       Generally, there is high variance in the detail of description between the different techniques

       Generally, it would be helpful to include efficiency values for all introduced methods (maybe also in the tables)

       The structure should be revised (more sections, see comments above)

Reviewer 3 Report

Cancers have been a burden and their early detection is crucial. The current study is tackling an interesting topic for the readers. Improving and advancing methods for detecting circulating tumor cells have always been a concern owing to specificity and sensitivity problems. Also, gathering information around the new technologies and studies help clinicians in their decisions. The manuscript is well-written, and the language is clear. Something that could be considered:

In the introduction section, the statistical data is based on 2020, I think it is better to update it with newer ones as this article will be published in 2023.

Round 2

Reviewer 2 Report

Manuscript ID: ijms-2127766

Title: Current and future methods for detecting circulating tumor cells

Vidlarova et al.

I would like to thank the authors for their visible efforts to improve the manuscript. The authors have addressed the main issues. The structure has overall improved significantly, but section 3 needs some further reworking (see comments). The table format has been improved and is now more readable. The introduction has been enriched with additional basic information about CTCs, making the following information much more understandable. Unnecessary methodological explanations have been removed and the depth of description for the single methods has mostly been balanced out. While the addition of clinical efficacy data is appreciated, some of it is not verifiable. Spiking experiments or comparisons of healthy versus patient data would be preferable. The focus on recent developments in CTC detection becomes clearer in the current manuscript and allows setting it apart from other reviews on the topic, however it still remains arguable how big the difference between the current and future techniques is on the methodological site. This aspect needs to be worked out more, as many of the future methods also rely on morphological or immunological features. The conclusion also lacks a clear summary, conclusion and outlook on the topic. Here it would also make sense to give a brief comparison of old versus new methodology.

In summary, I still recommend rejection review article.

Comments:

       Section 1. Introduction

â—¦       The introduction has been improved on all levels and now gives a good overview of the topic

       Section 2. Current enrichment and detection techniques:

â—¦       Section 2.1.1 is entitled affinity-based approaches, while the mentioned methods are referred to as immunology-based in table 1. I would suggest deciding on one term (preferably immunology-based)

â—¦       CellSearch is only mentioned very briefly even though it is the only FDA approved system. Mode of enrichment and detection are not described

â—¦       EpCAM-based is mentioned as a disadvantage multiple times, even for methods that don’t solely rely on EpCAM. Why? 

â—¦       2.2.1 Nucleic acid-based detection is too short. What markers are used? How are samples prepared? What about cell free DNA in the sample?

â—¦       Overalls description of the methods in 2.2 remains very brief and no efficiency data are provided

â—¦       Again need to check for accordance of subsection titles with table section titles

â—¦       Generally efficacy data from spiking or healthy/patient experiments would be preferable, as the detection of CTCs in patient samples only can hardly be verified

       Section 3. Future methods for CTC detection and characterization

â—¦       Even though the authors added an introduction to the section it is still not clear how the future methods are distinguished from the current ones. It is mentioned that viable cell isolation is a key concern, however this can also be achieved with some of the “old” methods. It seems the methods mainly differ in technology use and recency. Maybe the authors should consider renaming the section (and consequently changing the review title) to sth. Like recent advances in methods for detecting CTCs

â—¦       360-366: What is the significance of this section? No specific method is mentioned. Unclear.

â—¦       TetherChip principle is not clear from the text

â—¦       Section 3.6 remains very superficial. Also there is no “counterpart” about characterization in section 2. Why? 

â—¦       Generally the subsection titles are misleading. Most of the methods are still immunology or morphology based, it is irrelevant to the method whether the device is e.g. 3D printed 

â—¦       The section lacks a cohesive structure. Enrichment and detection methods are intermixed. Grouping of the methods is not always logical

       Section 4. Conclusion

â—¦       Again the conclusion remains very superficial, lacks a brief summary and fails to show the advantages of the future methods compared to the current ones, so that no real conclusion is drawn from the presented information
